# Sericin-Based Poly(Vinyl) Alcohol Relieves Plaque and Epidermal Lesions in Psoriasis; a Chance for Dressing Development in a Specific Area

**DOI:** 10.3390/ijms24010145

**Published:** 2022-12-21

**Authors:** Khwanchanok Tuentam, Pornanong Aramwit, Onrapak Reamtong, Suangsuda Supasai, Urai Chaisri, Kamonpan Fongsodsri, Rungnapha Yamdech, Napatara Tirawanchai, Passanesh Sukphopetch, Sumate Ampawong

**Affiliations:** 1Department of Tropical Pathology, Faculty of Tropical Medicine, Mahidol University, Ratchawithi Road, Ratchathewi, Bangkok 10400, Thailand; 2Center of Excellence in Bioactive Resources for Innovative Clinical Applications, Department of Pharmacy Practice, Faculty of Pharmaceutical Sciences, Chulalongkorn University, PhayaThai Road, Phatumwan, Bangkok 10330, Thailand; 3The Academy of Science, The Royal Society of Thailand, Dusit, Bangkok 10330, Thailand; 4Department of Molecular Tropical Medicine and Genetics, Faculty of Tropical Medicine, Mahidol University, Ratchawithi Road, Ratchathewi, Bangkok 10400, Thailand; 5Department of Biochemistry, Faculty of Medicine Siriraj Hospital, Mahidol University, 2 Prannok Road, Bangkok Noi, Bangkok 10700, Thailand; 6Department of Microbiology and Immunology, Faculty of Tropical Medicine, Mahidol University, Ratchawithi Road, Ratchathewi, Bangkok 10400, Thailand

**Keywords:** hydrogel, imiquimod, psoriasis, rat model, sericin

## Abstract

The noncontagious immune-mediated skin disease known as psoriasis is regarded as a chronic skin condition with a 0.09–11.4% global prevalence. The main obstacle to the eradication of the disease continues to be insufficient treatment options. Sericin, a natural biopolymer from *Bombyx mori* cocoons, can improve skin conditions via its immunomodulatory effect. Many external therapeutic methods are currently used to treat psoriasis, but sericin-based hydrogel is not yet used to treat plaques of eczema. Through the use of an imiquimod rat model, this study sought to identify the physical and chemical characteristics of a silk sericin-based poly(vinyl) alcohol (SS/PVA) hydrogel and assess both its therapeutic and toxic effects on psoriasis. The cytokines, chemokines, and genes involved in the pathogenesis of psoriasis were investigated, focusing on the immuno-pathological relationships. We discovered that the SS/PVA had a stable fabrication and proper release. Additionally, the anti-inflammatory, antioxidant, and anti-apoptotic properties of SS/PVA reduced the severity of psoriasis in both gross and microscopic skin lesions. This was demonstrated by a decrease in the epidermal histopathology score, upregulation of nuclear factor erythroid 2-related factor 2 and interleukin (IL)-10, and a decrease in the expression of tumor necrosis factor (TNF)-α and IL-20. Moreover, the genes S100a7a and S100a14 were downregulated. Additionally, in rats given the SS/PVA treatment, blood urea nitrogen, creatinine, and serum glutamic oxaloacetic transaminase levels were within normal limits. Our findings indicate that SS/PVA is safe and may be potentiated to treat psoriasis in a variety of forms and locations of plaque because of its physical, chemical, and biological characteristics.

## 1. Introduction

Psoriasis is a common T cell-mediated inflammatory skin condition that is non-contagious and characterized by excessively rapid keratinocyte proliferation. It primarily involves autoimmune reactions that are not appropriate and attack the skin’s target cells, especially keratinocytes and melanocytes, causing cellular hyperplasia and post-inflammatory hyperpigmentation, respectively [1]. Psoriasis is considered to be an incurable, chronic inflammatory skin condition with periods of improvement and worsening or intermittent symptoms. Furthermore, particularly in patients who present with generalized lesions, it is a highly prevalent skin condition that is closely related to mental illness [1,2]. The patients find it embarrassing to interact with people, especially young children. Topical, light, and systemic therapies are the three main types of psoriasis treatment [3]. Presently, topical therapy is produced from chemicals and natural extracts. Early results from chemical-based treatments have been positive, but long-term use by patients increases the risk of side effects. The opposite is also true: treatments made from natural extracts have become an alternative because they have a positive therapeutic impact and can be used in addition to other forms of therapy [4,5]. However, studies and research into the treatment of psoriasis are still needed to investigate and find new candidates to alleviate these symptoms.

The silk proteins sericin and fibroin, which are obtained from *Bombyx mori* cocoons, are significant sources of natural biopolymers [6]. Silk protein was once thought to be a waste product from the silk industry, produced during the grafting process after the cocoon had been boiled. When the silk protein cocoon is heated, the silk protein is washed off and discarded. However, numerous studies have shown that silk proteins have a wide range of beneficial properties, including the ability to be used as a medicine, dietary supplement, or ingredient in cosmetic products [6,7,8]. It also has anti-tyrosinase, which has antioxidative, anti-inflammatory, and moisturizing properties [6,9]. Interestingly, a recent study showed that cream-based sericin improved the psoriatic condition in animal models by modulating epidermal inflammation and decreasing epidermal thickening [10]. Accordingly, depending on the shape and severity of the psoriasis plaque, there is a high likelihood that sericin could be developed in a variety of forms for several topical therapeutic applications in psoriasis, such as creams, lotions, shampoos, and wound dressings. They would be suitable for patients to use in different plaque areas and in an appropriate way, leading to an effective treatment outcome.

Therefore, using a variety of techniques including histopathology, immunohistochemistry, and qRT-PCR, and based on the physical and chemical properties of sericin, this study investigated the effects of sericin-based poly(vinyl) alcohol (SS/PVA) on treatment for psoriasis in an imiquimod-induced rat model of psoriasis. The study provides a new alternative prototype for patients to have many choices of topical therapeutic applications and increase their quality of life. It also provides a foundation for future research and the search for alternative psoriasis treatments.

## 2. Results

### 2.1. Physical and Chemical Properties of SS/PVA

The percentages of the gel fraction in different fabricated formulas using the freeze–thaw technique are shown in Figure 1A. PVA and Dex/PVA, when compared with other formulations, had the highest percentage of the gel fraction (90.36% and 90.64%, respectively), while SS/PVA and Cal/PVA showed the greatest reduction in percentage (81.71% and 81.88%, respectively). However, this did not greatly affect the fabrication of the gel formation in the sheets. PVA sheets also had more solubility than the other formulas and kept its shape.

Biological degradation was assessed by the weight remaining, as shown in Figure 1B. The findings demonstrated that the degradation rates of Cal/PVA and SS/PVA (72.62% and 75.87%, respectively) were higher than those of PVA (82.45%) and Dex/PVA (89.20%) on the initial observation date. However, from Days 7 to 28, the residual weight in all the formulas did not show any significant differences. The degradation rates of Cal/PVA, SS/PVA, Dex/PVA, and PVA were 61.51%, 65.89%, 73.02, and 78.66%, respectively, after the study period.

The mechanical property was examined through the Young’s modulus, tensile strength, and elongation, as shown in Figure 1C–E. Young’s modulus and tensile strength were significantly higher in SS/PVA than in PVA (Figure 1C,D), while elongation showed no significant differences between the formulas (Figure 1E).

### 2.2. SS/PVA Reduced the Severity of Psoriatic Gross Skin Lesions in Imiquimod-Induced Psoriasis

The imiquimod-induced psoriasis rat model exhibited gross skin lesions as shown in Figure 2A(i–viii), including redness, scaling, and skin thickening. Redness and skin thickening were seen on Day 1 after induction and gradually got worse until Day 7 (Figure 2A(ii–viii)), whereas scaling was seen on Day 3 after induction and became worse on Day 5 (Figure 2A(vi–viii)). Regarding gross skin lesion scores, marked severity was first detected on Day 2 after induction and rapidly increased into a high score until the end of the induction period (Figure 2B). Gross skin lesion scores showed a significant decrease in all the treatment groups (Cal/PVA, Dex/PVA, and SS/PVA) after 7 days of treatment compared with the untreated group (Figure 2C). The outcome also showed that compared with the non-treated rats, the scores significantly declined in the rats treated with Dex/PVA, Cal/PVA, and SS/PVA on Days 1, 2, and 5 post-treatment, respectively (Figure 2D). Rats treated with Dex/PVA, Cal/PVA, and SS/PVA had significantly lower scores 1, 2, and 5 days after treatment, respectively, (Figure 2D).

### 2.3. SS/PVA Reduced Epidermal Thickness and Histopathological Scores in a Psoriatic Rat Model

The histopathological changes were examined to show that sericin can reduce microscopic skin lesions in the imiquimod-induced rat model of psoriasis. The main histopathological findings in the imiquimod-induced psoriatic rats consisted of acanthosis, dermatitis, folliculitis, and hyperkeratosis (Figure 3A(i–viii)). Figure 3Bi–ix shows the histopathological scores. When compared with the non-treated group, all treatment groups had significantly lower histopathological scores (Figure 3B(i)). However, pustule and dermatitis were highly prevalent in SS/PVA-treated rats when compared with other treatments (Figure 3B(ii,iii)). Epidermal hypoplasia, characterized by a very thin epidermal layer, was seen in rats treated with Dex/PVA (Figure 3B(iv)). After the study, rats that had received the Cal/PVA treatment also had acanthosis and squamous cysts (Figure 3B(v,vi)). Epidermal thickness in all treatment groups significantly decreased when compared with the non-treated group (Figure 3C(i–vi)).

### 2.4. SS/PVA Modulated Inflammation, Apoptosis, and Antioxidative Properties

Immunohistochemical analysis of the expression levels of IL-10, β-defensin, CCL-20, IL-10, IL-17, IL-20, IL-21, TGF-β, TNF-α Nrf-2, caspase-3, and caspase-9 on psoriatic skin models was carried out to clarify the effect of sericin on the modulation of inflammation, apoptosis, and antioxidant capacities. The results (Figure 4) showed that the expression of IL-10 in the SS/PVA-treated group significantly increased when compared with both the non-treated and all groups receiving the standard treatment. When compared with the control group, the expression of Nrf-2 was significantly higher in the SS/PVA and Dex/PVA treatment groups. When compared with the non-treated group, the expression levels of caspase-3 and caspase-9 in all treatment groups and the SS/PVA and Cal/PVA treatments were significantly reduced. In all treatment groups, there was a significant reduction in the expression of β-defensin. However, the expression of β-defensin in the Dex/PVA-treated group was significantly lower than in the SS/PVA-treated group. When compared with the untreated group, the expression levels of TNF-α and IL-20 in the SS/PVA and IL-20 in SS/PVA and Dex/PVA groups were significantly lower. The expression of CCL-20 in the Cal/PVA treatment group significantly increased when compared with the SS/PVA-treated and non-treated groups. The expression of CCL-20, however, tended to be lower in the SS/PVA-treated group. In comparison with all other treatments, the expression of TGF-β was significantly higher in the SS/PVA- and PVA-treated groups. Meanwhile, the expression of TGF-β tended to increase in the SS/PVA-treated and PVA-treated groups. In groups treated with SS/PVA and Dex/PVA, the expression levels of IL-17 and IL-21 tended to be reduced. However, there were no differences in the expression levels of IL-17 and IL-21 among any of the treatment groups.

### 2.5. Effect of Sericin on the Expression Levels of Psoriatic Genes in a Rat Model

Here, qRT-PCR was used to analyze the gene expression levels of FLG, caspase-14, involucrin, S100a7a, and S100a14 in psoriatic skin lesions with and without treatment. The results demonstrated that FLG (Figure 5A) and caspase-14 (Figure 5B) were significantly upregulated in skin specimens treated with Dex/PVA compared with Cal/PVA and SS/PVA. S100a7a was expressed more highly in the Cal/PVA-treated group than in the non-treated group (Figure 5C), whereas the Dex/PVA- and SS/PVA-treated groups had lower levels of expression. Furthermore, compared with the Cal/PVA-treated group, the expression of S100a14 was upregulated in the Cal/PVA-treated group and significantly downregulated in the SS/PVA- and Dex/PVA-treated groups (Figure 5D). However, the expression of involucrin was not detected in any group of psoriatic rat skin specimens.

### 2.6. Complete Blood Count and Clinical Blood Chemistry

To evaluate the toxicological effect of the treatments, complete blood counts and clinical blood chemistry were examined. When compared with the normal range, the imiquimod-induced psoriasis rat model showed a decrease in red blood cells (RBC) and hemoglobin (HGB), and an increase in white blood cells (WBC) (Table 1). A significant difference was observed in WBC and RBC between non-treated and Dex/PVA groups, and in HGB between the Dex/PVA and SS/PVA treatments. All groups showed an increase in neutrophils, eosinophils, and basophils; however, the eosinophil count in the sericin-treated group was significantly lower than that in the Dex/PVA-treated group. Monocyte and lymphocyte numbers fell in all groups, but lymphocyte numbers in the Dex/PVA-treated group were significantly lower than expected. In addition, liver function was evaluated by the liver enzymes. The levels of serum glutamate-pyruvate transaminase (SGPT) and glutamic oxaloacetic transaminase (SGOT) in all groups were within normal ranges. In addition, renal function was evaluated by BUN and creatinine levels; the results indicated that all groups were in the normal ranges.

## 3. Discussion

Our study aimed to explore the physical, chemical, therapeutic, and toxic properties of SS/PVA in psoriasis using an imiquimod rat model. Our overall findings are summarized in Figure 6. The results showed that SS/PVA had some positive effects on psoriasis, as shown by relief from severe skin lesions (scaling, redness, and thickness) and a decrease in epidermal lesions (acanthosis, dermatitis, folliculitis, and hyperkeratosis) when it was manufactured and released properly, which it achieved by increasing IL-10 and Nrf-2 and decreasing TNF-α, IL-20, β-defensin, caspase-3, and caspase-9. Sericin also demonstrated anti-inflammatory, antioxidant, and anti-apoptotic activities. Sericin decreased the expression of the S100a7a and S100a14 genes, which are involved in the severity of psoriasis. Moreover, SS/PVA produced a non-toxic effect, as characterized by some liver and kidney enzymes, particularly SGOT, BUN, and creatinine.

Hydrogels are mainly used to construct wound dressing materials to keep the wound moist and allow metabolites to pass freely [11]. Additionally, sericin-based hydrogels have excellent biocompatibility and can load and release small molecules [12]. In this study, a hydrogel dressing was created using PVA and various substrate concentrations, including 4% sericin, 0.1% dexamethasone, and 30 μg/μL calcitriol. Regarding the chemical properties (Figure 1A), SS/PVA and Cal/PVA PVA, and Dex/PVA were shown to have a lower percentage of the gel fraction than Dex/PVA, in agreement with Aramwit et al., who suggested that the concentration of the substrate had a negative effect on the percentage of the gel fraction [13]. Furthermore, the physical characteristics of all dressings were assessed in terms of Young’s modulus, tensile strength, and elongation capacity (Figure 1C–E). All dressing materials had strongly and flexible attachment properties, especially SS/PVA, which had a greater elongation factor than the others. As a result, the chemical and physical advantages of SS/PVA may have a strong ability to adhere to the skin’s surface, particularly in various plaque areas in psoriatic patients.

The pathogenesis of psoriasis is related to uncontrolled keratinocyte proliferation and differentiation due to the dysregulation of the immunological cells’ function [2,14]. Hyperkeratosis, epidermal acanthosis, dilated vasculature, and leukocytic infiltrate are all visible in psoriatic plaque’s histopathology [15]. According to our findings, all treatments effectively reduced the amount of gross psoriatic skin when compared with the non-treated group (Figure 2C,D). However, the histopathological score gave different results, namely that Dex/PVA induced epidermal hypoplasia, which presented as epidermal atrophy (Figure 3(Bix)). According to a different study, topical steroids have been linked to a variety of side effects, including skin atrophy, striae distensae, folliculitis, and purpura [16]. These may be a contraindication to the long-term use of steroids for psoriasis. In addition, acanthosis and squamous cysts were still present in the rats treated with Cal/PVA (Figure 3B(v,vi)), while they were not observed in SS/PVA- and Dex/PVA-treated rats. The pathological findings in sericin still demonstrated pustules and dermatitis when compared with another treatment (Figure 3B(ii,iii)). Regarding the histopathological results, SS/PVA had more beneficial effects at the epidermal level than on deeper skin lesions. Therefore, any treatment has its own either benefit or limitations for treating psoriasis, depending on the duration of use, the concentration, the formulation, and the related properties. According to our most recent study, we discovered that 10% sericin in a cream was the most effective dose for treating psoriasis [10]. There are currently several topical therapy treatments available, each with unique benefits. Coal tar and tazarotene can suppress epidermal proliferation and inflammation, and can reduce plaque-like psoriatic lesions [17,18,19]. In psoriatic skin, dithranol can lessen epidermal hyperproliferation and inflammatory infiltrate [20]. Through the TLR-VDR pathways, Vitamin D (analog) can increase the effectiveness of UVB phototherapy in psoriasis without causing negative side effects on the skin [21].

An abnormal T cell-mediated immune response is considered a major cause of psoriasis [1,2]. Presently, there are three stages to the pathogenesis linked to the expression of cytokines and chemokines: the innate immune response, the adaptive immune response, and the auto-amplification loop [1]. In the initial step, keratinocytes, as the skin barrier, secrete the antimicrobial peptides (AMP) such as LL37, β-defensin, and S100. When AMP binds to the plasmacytoid dendritic cells (TLR7) and TLR9, it binds to the myeloid dendritic TLR8 cells and causes them to produce many Type I interferons such as interferon-gamma (IFN-γ), IL-6, and IL-β to activate mature dendritic cells [2]. The production of TNF-α, IL-12, IL-23, and IL-6 from the mature dendritic cells can modulate T cells to differentiation and proliferation [22,23]. The pro-inflammatory cytokines IL-17a, IL-22, IL-21, TNF, and IFN are secreted by T-helper 1 (Th1), T-helper 17 (Th17), and T-helper 22 (Th22) cells when T-cells are activated. These cytokines can cause dendritic cell activation, proliferation, and epidermal inflammation in keratinocytes [1,2]. Moreover, reactive oxygen species in psoriasis involve the regulation of the oxidoreductive balance of oxidative stress, which can lead to inflammation and apoptosis [24,25,26]. SS/PVA and Dex/PVA demonstrated an antioxidant capacity through Nrf-2 as our study progressed (Figure 4). When compared with the non-treated group, SS/PVA increased the anti-inflammatory cytokine IL-10, which can lower pro-inflammatory cytokines such as β-defensin, TNF-α, and IL-20. However, under the SS/PVA treatment, CCL-20, IL-17, and IL-21 tended to decrease. Therefore, the main properties of SS/PVA are its anti-inflammatory and anti-oxidative effects from the upregulation of IL-10 and Nrf-2 and the downregulation of β-defensin, TNF-α, and IL-20. Natural remedies such as curcumin and anthocyanin have been created to treat psoriasis in various ways. By reducing the levels of pro-inflammatory cytokines such as TNF-α, IFN-γ, IL-2, IL-17A, IL-17F, IL-22, and IL-23, curcumin extract can reduce inflammation [27,28,29]. Anthocyanin can reduce the levels of pro-inflammatory cytokines (IL-6, IL-8, IL-20, IL-22, and TNF-α), chemokines (CCL-20), and antimicrobial peptides (β-defensin), and enhance the antioxidative properties (Nrf-2) [1]. In terms of the dysfunctional apoptosis brought on by psoriasis, the expression of caspase-9 in the SS/PVA and Cal/PVA groups was significantly lower than that seen in the untreated group. Additionally, when compared with no treatment, the expression of caspase-3 was significantly reduced in all treatment groups. Apoptotic dysfunction is considered to play a major role in several skin diseases, particularly psoriasis [30], leading to excessive proliferation and abnormal differentiation of the keratinocytes [31]. In the current study, the SS/PVA and Cal/PVA treatments were able to decrease acanthosis, hyperkeratosis, and epidermal thickening, in conjunction with the downregulation of caspase-3 and -9. Bears et al. reported that the overexpression of caspase-3 plays a pivotal role in the pathogenesis of psoriasis and is related to the onset of psoriatic skin lesions [32]. These findings showed that sericin inhibits keratinocyte proliferation, which results not only from its anti-inflammatory and antioxidative properties but also from the effect of regulating epidermal apoptosis.

The current study evaluated the toxicological effects of sericin using hematological parameters and blood clinical chemistry in addition to examining its physical, chemical, and therapeutic properties. Imiquimod-induced psoriasis in rats closely resembles plaque-type psoriasis in humans, which is characterized by the infiltration of inflammatory cells into the epidermis, including T-cells, neutrophils, and dendritic cells [33], which affect hematological function. Our hematological results indicated a mild decrease in RBC and HGB in all groups. All groups, particularly those with neutrophilia, basophilia, and eosinophilia, also had leukocytosis. The recruitment of neutrophils and eosinophils is greatly influenced by mast cells, a resident form of basophil tissue, and the increase in neutrophils may be related to some triggering of these cells [34,35,36]. Neutrophils initiate immune reactions involving T cell imbalance, keratinocyte proliferation, angiogenesis, and auto-antigen formation [37]. In psoriatic skin lesions, neutrophils are also regarded as a pre-activated cell [38]. Furthermore, the degree of severity in psoriasis is directly correlated with neutrophilia [39,40]. Unfortunately, no amount of treatment was able to lower the neutrophil counts, especially seven days after treatment began. Eosinophilia might be attributed to concurrent allergic or atopic dermatitis induced by the application of dexamethasone [41]. In contrast to the Dex/PVA-treated and untreated rats, our study showed that the SS/PVA treatment reduced the level of eosinophils in circulating blood. Additionally, rats in the Dex/PVA group had significantly fewer total lymphocytes than the other groups, which may be related to the immuno-suppressive effects of steroids [42]. These findings indicated the adverse effect of steroids. Clinical blood chemistry revealed that all treatments had normal renal blood profiles, but all treatments had higher SGPT levels than would be expected. However, compared with the Cal/PVA and Dex/PVA groups, the rats treated with SS/PVA had significantly lower levels of SGPT and SGOT. This may be a contraindication for the use of sericin, calcitriol, and dexamethasone for treating psoriasis, especially at high concentrations or with long-term usage.

To evaluate the specific genes involved in the pathogenesis of psoriasis, qRT-PCR was performed. Caspase-14, FLG, and involucrin maintain epidermal integrity both in terms of structure and function [43,44]. Inflamed psoriatic skin contains psoriasin, which cause epidermal necrosis and degeneration [44]. In this study, qRT-PCR indicated that Dex/PVA-treated rats showed an upregulation of the mRNA expression levels of caspase-14 and filaggrin (FLG) when compared with the Cal/PVA treatment (Figure 5A,B). S100a7a and S100a14, two genes associated with the severity psoriasis, showed decreased expression levels in the SS/PVA-treated group (Figure 5C,D), which led to an improvement in the psoriatic condition. However, the expression of involucrin was not detected in this experiment. Involucrin is also only found in human psoriasis [44,45]. There was a rumor that this gene might work well in the rat model. 

According to our chemical, physical, therapeutic, and toxic analysis of SS/PVA, there may be some possible limitations in this study. Firstly, our results might not reflect the full capacity of sericin for treating psoriasis due the short period of the study. Some specific indicators of recovery such as serum chemokines and cytokines were not explored in this study. Materials were prepared as prototypes and were used at a laboratory scale. Lastly, the psoriatic plaque induction areas on the rats were limited to the dorsal region.

## 4. Materials and Methods

### 4.1. Sericin Extraction

*Bombyx mori* cocoons were purchased from Chul Thai Silk Co., Ltd., Phetchabun, Thailand. Fresh cocoon shells were autoclaved in distilled water for 1 h at 120 °C to extract the sericin [46]. The sericin-containing supernatant was gathered, filtered, and stored in a desiccator. Every batch of sericin extract was examined for amino acid composition as a quality control, which was performed by Central Laboratory (Thailand) Co., Ltd., Bangkok, Thailand.

### 4.2. Hydrogel Preparation

A hydrogel-based formulation [13] was used to prepare the plaque dressings, which were divided into 4 groups: (i) pure hydrogel (PVA), (ii) 4% sericin (SS/PVA), (iii) 3 µg/mL calcitriol (Cal/PVA), and (iv) 0.1% dexamethasone (Dex/PVA). Mixtures of 6% PVA with 9.6 mL sericin, 12 g of calcitriol, and 0.2 g of dexamethasone were prepared. They were combined using a magnetic stirrer to ensure homogeneity, then the mixture was then fabricated to form a hydrogel by the freeze–thaw method. Briefly, it was poured into a mold and stored at −20 °C for 24 h before being brought to room temperature for 6 h; this process was repeated 4 times. The hydrogel was sliced by a cutting machine, trimmed into 1 × 2 cm^2^ pieces, and sterilized by UV radiation before being used in the experiment.

### 4.3. Physical and Chemical Properties

The physical and chemical properties of the hydrogels were evaluated following the methods reported by Yamdej et al. [47]. The hydrogels were prepared for testing by cutting them into pieces measuring 1 × 2 cm^2^ and comparing the dried weight of the hydrogel with the dry weight of the hydrogel that was still present after being submerged in deionized water. The hydrogel was placed in a phosphate-buffer saline solution (PBS, pH 7.4) for an in vitro enzymatic biodegradation test, and the PBS was collected at various time points for measuring the protein released using a BCA protein assay kit. The compressive and tensile moduli of the hydrogels were calculated according to the ASTM D638-01 method after a mechanical test was conducted on the hydrogel in each substrate using a Universal Testing Machine (Hounsfield H10KM, UK) fitted with a 10 kN load cell at a constant rate of 30 mm/min.

### 4.4. Animal Experimental Protocol

#### 4.4.1. Animal Ethics Statement

The Faculty of Tropical Medicine ACUC a Mahidol University was asked for approval for the animal experiments (Approval No. FTM-ACUC 023/2021). The National Research Council of Thailand’s Guidelines for the Use of Animals and the Thai Animals for Scientific Purposes Act, both passed in B.E. 2558, were put into effect. Twenty 8-week-old female Wistar rats were obtained from the Nomura-Siam International company, Thailand. They were kept in ventilated and humidity-controlled spaces with a 12-h/12-h light/dark cycle, and they were given access to a standard diet ad libitum. Additionally, the animals were accustomed to the new environment (acclimatization) for 7 days before the induction period.

#### 4.4.2. Induction of Psoriasis and Experimental Protocol

To induce psoriasis in the rat model, 62.5 mg of imiquimod [48] was applied to the shaved dorsal skin of rats, limited to a 3 cm^2^ area for 7 days (induction period) once a day. Scaling, redness, and skin thickening were the three parameters that made up the severity score, which were graded on a scale of 0–4 (0 = absent, 1 = mild, 2 = moderate, 3 = severe, and 4 = very severe). The experiment was separated into 4 groups, each group consisting of 5 experimental animals. PVA, SS/PVA, Cal/PVA, and Dex/PVA were each applied separately to the rats in each group. To keep maintain the psoriatic skin condition during the treatment in all groups of the experiment, imiquimod was continuously administered to the rats after the induction period. All of the test materials were applied daily to shaved dorsal skin for 7 days after the application of imiquimod for 1 h.

### 4.5. Sample Collection

All rats were humanely euthanized by an overdose of carbon dioxide inhalation. Blood was collected from the rats via heart puncture for hematological and blood clinical chemistry testing, which was carried out by the National Laboratory Animal Center, Mahidol University. Skin lesions from psoriasis were autopsied. They were preserved in 10% neutral buffer formalin for histopathological and immunohistochemical studies. For molecular analysis via qRT-PCR, some of them were kept in an RNAlater stabilization solution.

### 4.6. Histopathological and Immunohistochemistry Studies

#### 4.6.1. Histopathological Examination

To assess the severity of the histopathological changes in the skins of psoriatic rats, the standard tissue processing steps of dehydration with an ethanol gradient (2 min of each; 30%, 50%, 70%, 90%, and 100%), infiltration with a xylene gradient (15 min of each), and paraffin embedding were used to prepare the fixed specimens. The sections were deparaffinized in xylene (twice for 10 min), hydrated in a graded series of ethanol (2 min each; 90% and 70%), and stained with hematoxylin and eosin after being cut to a thickness of 5 nm (H&E). The sections were examined under a light microscope (BX41, Olympus, Shinjuku, Tokyo, Japan) and color images were acquired by a digital camera (DP20, Olympus, Shinjuku, Tokyo, Japan) at 400× magnification. ImageJ software (version 1.36) was used to measure the thickness of the epidermis (NIH, Bethesda, MD, USA). The presence of hyperkeratosis, acanthosis, folliculitis, and dermatitis on the psoriatic skin in the untreated, SS/PVA-treated, Cal/PVA-treated, and Dex/PVA-treated groups was used to determine the level of hyperkeratosis. The histopathological changes were scored using the H-score. The severity score (0–3; 0 = absent, 1 = mild, 2 = moderate, and 3 = severe) and extent of distribution (0–100%) were multiplied to produce the H-score (0–300).

#### 4.6.2. Immunohistochemistry

To demonstrate the anti-psoriatic properties of sericin, rabbit isotype polyclonal antibodies (MyBioSource, San Diego, CA, USA) were used as the primary antibodies. These antibodies target caspase-3, caspase-9, nuclear factor erythroid 2-related factor 2 (Nrf-2), interleukin (IL)-6, IL-10, IL-17, IL-20, IL-21, IL-22, C-C motif chemokine ligand 20 (CCL20), β-defensin, tumor necrotic factor (TNF)-α, and transforming growth factor (TGF)-β. After deparaffinization with xylene (twice for 10 min) and hydration, the antigenicity of the sections was enhanced by heat-induced antigen retrieval in a citrate buffer (pH 6) using a microwave method twice for 8 min, followed by endogenous peroxidase blocking with 1% hydrogen peroxide in methanol for 10 min, and non-specific binding in EnVision FLEX/horseradish peroxidase (HRP) blocking reagent (DAKO, Santa Clara, CA, USA) for 10 min. After incubation with the primary antibodies for 1 h, the sections were incubated with polymer HRP anti-mouse/rabbit (DAKO) labeling for 30 min, and diaminobenzidine (DAKO, Santa Clara, CA, USA) visualization was performed on the sections for 3 min. The sections were examined under a light microscope (BX41, Olympus, Shinjuku, Tokyo, Japan) after being counterstained with hematoxylin and mounted with DEPEX (Electron Microscopy Sciences, Hatfield, PA, USA).

A light microscope equipped with a digital camera and at least five fields per section was used to take color images of each skin sample with a resolution of 640 × 480 pixels (400× magnification) to measure the expression levels of all marker proteins. The image analysis program was used to localize the areas of expression, and the H-score (percentage area of expression × intensity score) was calculated to determine the immunological labeling. The immunolabeled area was adjusted using the threshold mode, and the images were converted to grayscale. Measurements were made of the expression areas (percentage). Using a grading scale with four levels (0–3; 0 = negative staining, 1 = low-intensity staining, 2 = moderate-intensity staining, and 3 = high-intensity staining) the levels of immunolabeling intensity were evaluated.

### 4.7. Quantitative Real-Time Polymerase Chain Reaction (qRT-PCR)

To measure the specific genes involved in the pathogenesis of psoriasis, the expression levels of several genes, such as FLG, caspase-14, involucrin, S100a7a, and S100a14, were examined by qRT-PCR [44].

#### 4.7.1. RNA Extraction

The rat skin samples preserved in RNAlater were extracted using the protocol as recommended by the manufacturing company of the total RNA purification kit (RNeasy Mini Kit Qiagen, Toronto, Ontario, Canada). Rat skins were briefly homogenized in a lysis buffer, and the lysate was loaded into a spin column. The eluent was discarded after the RNA had been bound to the column and washed with a buffer. The RNA was eluted in RNase-free water, and the RNA concentrations were measured by a NanoDrop™ 2000/2000c spectrophotometer (Thermo Scientific, Waltham, MA, USA).

#### 4.7.2. qRT-PCR

qRT-PCR was performed by using the iTaq™ Universal SYBR Green Supermix (BIO-RAD, Hercules, CA, USA). In the CFX96 Touch Real-time PCR detector, the primer pairs were used as shown in Table 2 for amplification and denaturation (BIO-RAD, Hercules, CA, USA). By using the 2^−∆∆Ct^ method, the individual gene expression levels were calculated. The expression level of β-Actin was used as a stable reference gene for accurate normalization of the gene expression data.

### 4.8. Statistical Analysis

Data were expressed as the mean and standard error of the mean during the statistical analysis, which was carried out using statistical analysis software (IBM SPSS, version 18 and GraphPad Prism, version 9, Armonk, NY, USA); *p* < 0.05 was used to indicate statistical significance. The Kolmogorov–Smirnov test was used to determine the distribution of the data. To compare the differences among the groups in parameters such as the instant histopathological score (H-score), immunohistochemical labeling (H-score), the physical and chemical properties, clinical blood chemistry, complete blood counts, and the relative mRNA expression levels, parametric and non-parametric *t*-tests or analysis of variance (ANOVA) were chosen as appropriate. The significant level in the figure was presented as * = *p* < 0.05, ** = *p* < 0.01, *** = *p* < 0.001, **** = *p* < 0.0001.

## 5. Conclusions

In conclusion, an effective dressing technique offers a high likelihood of use in conjunction with chemical medications or natural active ingredients for application in psoriatic lesions of various forms and sizes. This study provides some evidence of the benefits of sericin-based poly(vinyl) alcohol for the alleviation of psoriasis, especially in terms of the immunohistopathological, molecular, and toxicological aspects. However, three other main aspects should be examined in further studies: (i) the therapeutic mechanisms of sericin in treating psoriasis should be clarify which genes or proteins are involved in the recovery effect using next-generation sequencing or proteomic studies; (ii) cost-effectiveness should be studied regarding how to scale up from the laboratory scale to a commercial prototype; (iii) improving the performance of sericin purity should be explored, including which sections of its protein sequences (e.g., sericin 1, sericin 2, and other elements) play important roles in the anti-psoriatic properties. Achieving a better therapeutic result through our present and additional studies would be helpful for future development of dressings and clinical studies.

## Figures and Tables

**Figure 1 ijms-24-00145-f001:**
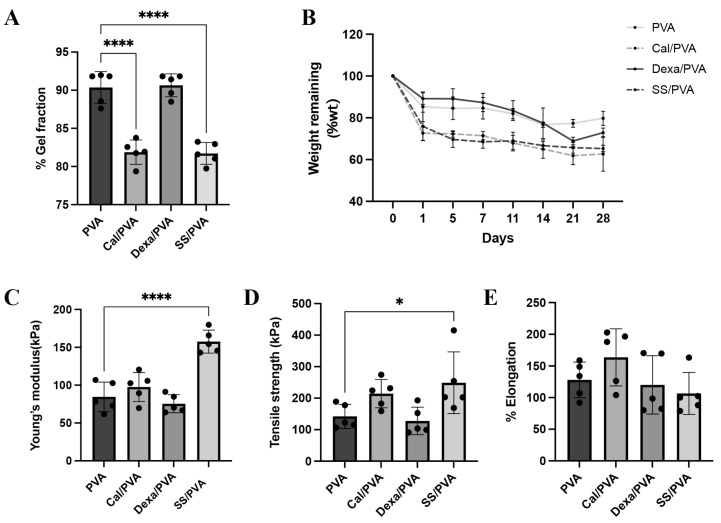
Physical and chemical properties of the test materials: (**A**) Bar graph of the evaluated gel fraction of hydrogel (**B**) Line graph of weight remaining after in vitro biodegradation of the hydrogel (**C**–**E**) Bar graphs of the mechanical properties of the hydrogel under tensile testing: (**C**) Young’s modulus, (**D**) tensile strength, and (**E**) elongation. The significant level in the figure was presented as * = *p* < 0.05, **** = *p* < 0.0001.

**Figure 2 ijms-24-00145-f002:**
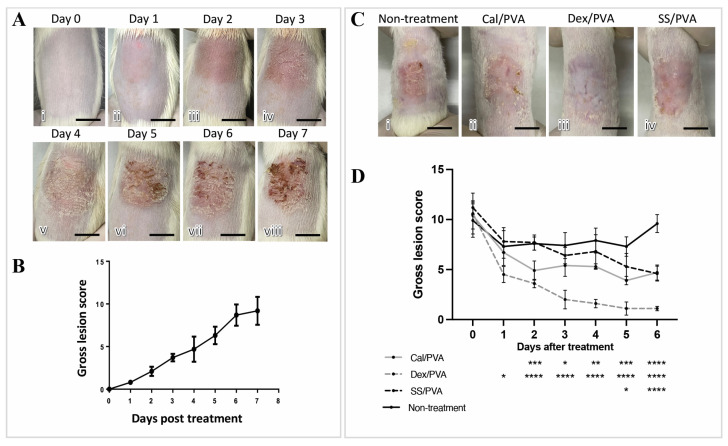
Anatomical appearance scores after treatment in imiquimod-induced psoriatic rat models. (**A**) Gross skin lesions in the induction period from Day 0 to Day 7 (**i**–**viii**). (**B**) Line graphs of the severity scores of psoriatic lesions in the induction period from Day 0 to Day 7. (**C**) Gross skin lesions in each group after treatment on Day 6 (**i**–**vi**): (**i**) untreated; (**ii**) calcitriol-based hydrogel treatment; (**iii**) dexamethasone-based hydrogel treatment; (**vi**) sericin-based hydrogel treatment. (**D**) Line graph comparing gross lesion scores after treatment from Day 0 to Day 6 in each group, with the significance level of treated groups compared with the untreated control determined using ANOVA, scale bars (**A**,**C**) = 0.75 cm. The significant level in the figure was presented as * = *p* < 0.05, ** = *p* < 0.01, *** = *p* < 0.001, **** = *p* < 0.0001.

**Figure 3 ijms-24-00145-f003:**
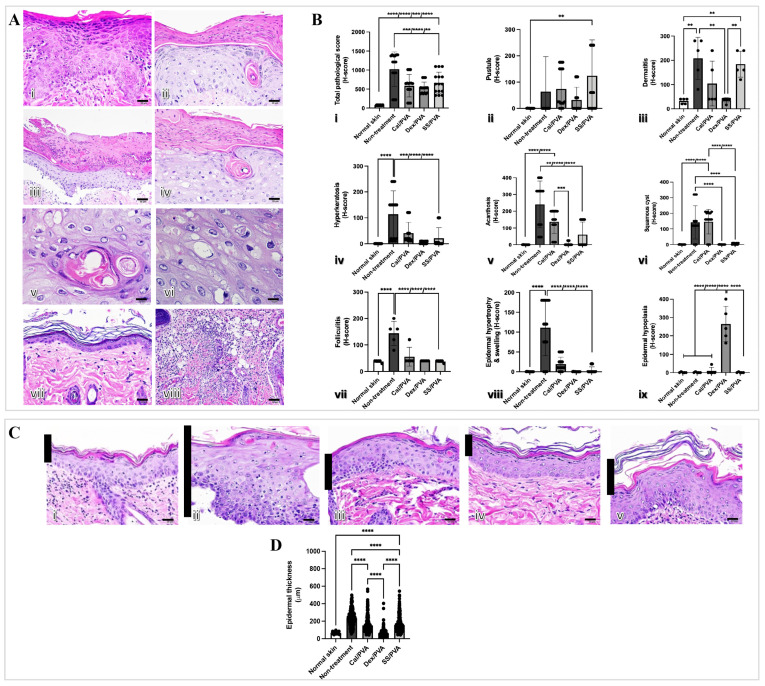
Epidermal thickness and histopathological evaluation of imiquimod-induced psoriatic rat skins after treatment. (**A**) Histological appearances in imiquimod-induced psoriatic rats after 7 days of induction (**i**–**viii**): (**i**) acanthosis; (**ii**) hyperkeratosis; (**iii**) pustules; (**iv**) epidermal hypertrophy with cysts; (**v**) squamous cyst (high magnification); (**vi**) epidermal hypertrophy (high magnification); (**vii**) epidermal hypoplasia; (**viii**) folliculitis with dermatitis. (**B**) Bar graph of pathological scores involving psoriasis derived using ANOVA (**i**–**ix**): (**i**) total pathological score; (**ii**) pustules; (**iii**) dermatitis; (**iv**) hyperkeratosis; (**v**) acanthosis; (**vi**) squamous cysts; (**vii**) folliculitis; (**viii**) epidermal hypertrophy and swelling; (**ix**) epidermal hypoplasia. (**C**) Epidermal thickness stained with hematoxylin and eosin (H&E) on Day 7 of post-treatment in the psoriatic skin of rats (**i**–**v**) at 400× magnification: (**i**) normal skin; (**ii**) no treatment; (**iii**) calcitriol; (**iv**) dexamethasone; (**v**) sericin. (**D**) Bar graph of total epidermal thickness derived using ANOVA. Scale bars (**A**(**i**–**iv**,**vii**,**viii**),**C**(**i**–**v**)) = 20 μm, (**A**(**v**,**vi**)) = 10 μm. The significant level in the figure was presented as ** = *p* < 0.01, *** = *p* < 0.001, **** = *p* < 0.0001.

**Figure 4 ijms-24-00145-f004:**
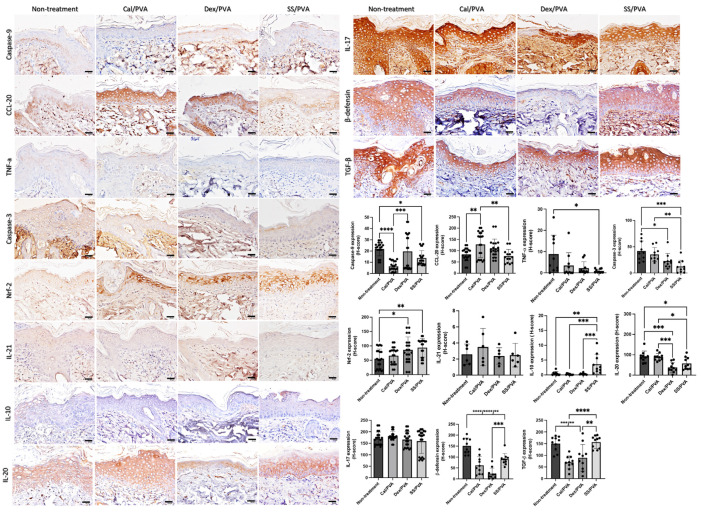
The expression of cytokines and chemokines involved in psoriatic skin determined using immunohistochemical staining**.** Bar graph of the H-scores using ANOVA; scale bars = 20 μm. The significant level in the figure was presented as * = *p* < 0.05, ** = *p* < 0.01, *** = *p* < 0.001, **** = *p* < 0.0001.

**Figure 5 ijms-24-00145-f005:**
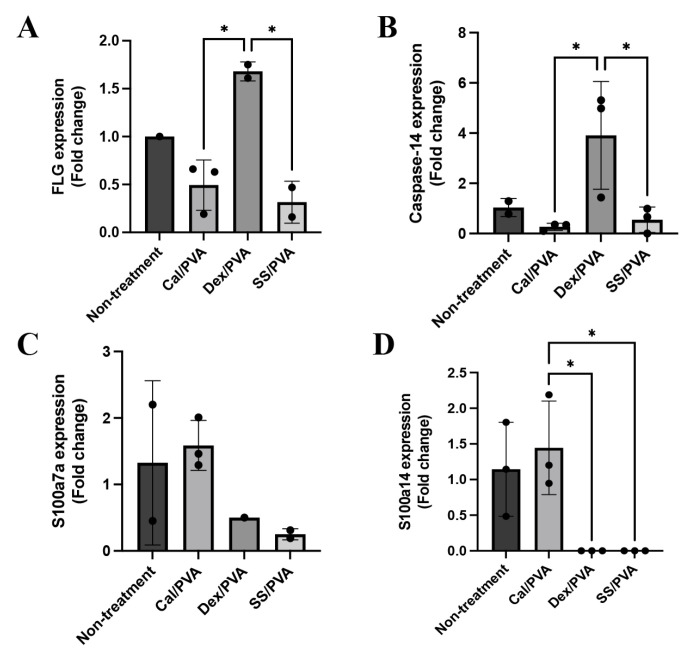
Psoriatic gene expression levels in a rat model. (**A**) Bar graph of relative mRNA FLG expression levels in rat skin specimens. (**B**) Bar graph of relative mRNA caspase-14 expression levels in rat skin specimens. (**C**) Bar graph of relative mRNA S100a7a expression levels in rat skin specimens. (**D**) Bar graph of relative mRNA S100a14 expression levels in rat skin specimens. The significant level in the figure was presented as * = *p* < 0.05.

**Figure 6 ijms-24-00145-f006:**
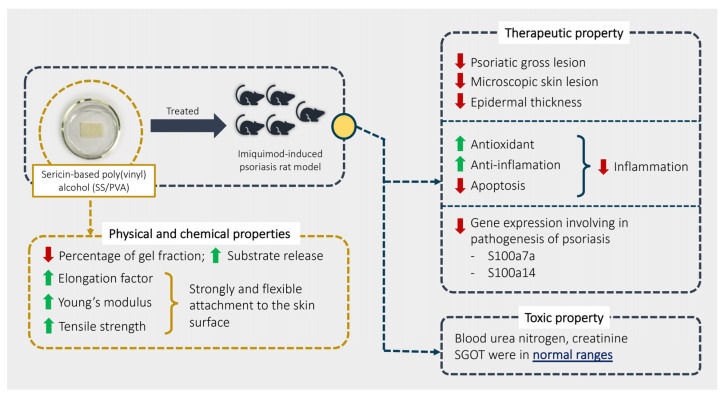
Summary of the physical and chemical properties of SS/PVA and its therapeutic and toxic effects.

**Table 1 ijms-24-00145-t001:** Complete blood count and chemical chemistry (mean ± S.D).

Parameter	Complete Blood Count and Blood Chemical Chemistry (Mean ± S.D)				
	Standard	Hydrogel	Calcitriol	Dexamethasone	Sericin
WBC (10^6^/µL)	4.23 ± 0.72	9.19 ± 4.30	8.25 ± 1.51 *	2.88 ± 1.64 *	7.71 ± 3.80
RBC (10^6^/µL)	9.27 ± 0.63	7.27 ± 0.25 *	7.52 ± 0.50	7.92 ± 0.25 *	7.31 ± 0.36
HGB (g/dL)	17.78 ± 1.02	13.76 ± 0.58	14.18 ±0.86	14.56 ± 0.55*	13.58 ± 0.61 *
HCT (%)	56.45 ± 3.80	41.40 ± 1.81	42.30 ± 2.70	44.40 ± 2.41	40.92 ± 1.48
MCV (fl)	60.93 ± 1.75	56.94 ± 1.65	56.26 ± 2.26	56.00 ± 2.07	55.98 ± 1.04
MCH (pg)	19.20 ± 0.51	18.94 ± 0.49	18.86 ±0.58	18.88 ± 0.43	18.58 ± 0.33
MCHC (g/dL)	31.50 ± 0.38	33.24 ± 0.55	33.50 ± 0.51	33.72 ± 0.71	33.18 ± 0.31
PLT (10^3^/µL)	804.50 ± 136.88	642.00 ± 54.12 *	728.60 ± 123.32	473.00 ± 90.12 *	631.40 ± 195.39
RDW (%)	17.79 ± 1.94	29.30 ± 1.14	27.82 ± 1.53	30.66 ± 1.52	28.26 ± 0.72
PDW (fl)	8.41 ± 0.45	8.62 ± 0.31	8.88 ± 0.33	8.26 ± 0.82	8.72 ± 0.48
RET (K/µL)	232.88 ± 53.78	411.04 ± 33.64	450.92 ± 119.98	327.36 ± 75.09	404.38 ± 40.20
MPV (fl)	7.63 ± 0.69	8.96 ± 0.29	8.74 ± 0.58	8.44 ± 1.03	8.84 ± 0.54
PCT (%)	0.61 ± 0.08	0.57 ± 0.06 *	0.62 ± 0.14	0.39 ± 0.08 *	0.56 ± 0.20
Neutrophils (%)	11.73 ± 6.98	15.90 ± 9.53	10.90 ± 11.06	13.00 ± 18.58	16.90 ± 5.51
Lymphocytes (%)	79.88 ± 7.03	72.42 ± 6.16	74.82 ± 15.05	60.60 ± 14.83	72.68 ± 5.50
Eosinophils (%)	1.17 ± 0.33	10.00 ± 6.30	12.08 ± 7.16	25.22 ± 8.14 *	8.32 ± 1.74 *
Basophils (%)	0.35 ± 0.38	1.62 ± 0.97	2.04 ± 1.35	1.02 ± 0.60	2.02 ± 0.82
Monocytes (%)	6.87 ± 1.49	0.06 ± 0.05	0.16 ± 0.25	0.16 ± 0.23	0.08 ± 0.08
Blood urea nitrogen (mg/dL)	20.73 ± 2.51	10.98 ± 5.74	15.84 ± 5.50	22.28 ± 5.36	23.34 ± 5.99
Creatinine (mg/dL)	0.67 ± 0.04	0.07 ± 0.07	0.14 ± 0.08	0.16 ± 0.05	0.21 ± 0.06
SGPT (U/L)	53.18 ± 10.15	81.86 ± 34.63	265.56 ± 329.42	475.98 ± 688.57	202.12 ± 166.64
SGOT (U/L)	93.73 ± 11.96	21.06 ± 14.07	49.58 ± 35.48	55.20 ± 46.14	47.30 ± 16.67

* Matched difference between groups.

**Table 2 ijms-24-00145-t002:** Primers used for qRT-PCR.

Gene	Primers	
FLG	F	5′AGATGTGGACCACGATGACAA3′
	R	5′TAGTGCTGGATCCTCGTCTTTT3′
β-Actin	F	5′CACTATCGGCAATGAGCGGTTCC3′
	R	5′AGCACTGTGTTGGCATAGAGGTC3′
Caspase-14	F	5′CAGACCCTGACGGATGTGTTC3′
	R	5′GCGAGGGTGCTTTGGATTTCGG3′
Involucrin	F	5′TGTAGGGGTTTGCTGCGTAAG3′
	R	5′AGTCACTGGCACTGTGTGTTG3′
S100a7a	F	5′TAGTGTGCCTCGCTTCATGGAC3′
	R	5′CACAACTGCCGGTGAAACTGA3′
S100a14	F	5′AACAATGGGACAGTGTCGGTC3′
	R	5′ACTGCTGGGTAACCAGGTCTC3′

## Data Availability

Not applicable.

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
