# Peer review of "Sericin-Based Poly(Vinyl) Alcohol Relieves Plaque and Epidermal Lesions in Psoriasis; a Chance for Dressing Development in a Specific Area"

_ijms, 2022, doi:10.3390/ijms24010145_

Round 1

Reviewer 1 Report

Tuentam et al. reported the design and fabrication of silk-sericin based hydrogel composite for the treatment of skin psoriasis.  

This is thorough, well done, experimental work and I congratulate the authors on their nice study. Nevertheless, I am still of the view that this work lacks the high degree of novelty required

This is thorough, well done, experimental work and I congratulate the authors on their nice study. Nevertheless, there are some gaps related figure 2 and 3. Scale bar was missing and also statistic report are not very clear. What type of statistical analysis was done?

Also there are gaps related the cell type description of these figure.

Reviewer 2 Report

This report (ID: ijms-2018943) entitled “Sericin-based poly(vinyl) alcohol relieves plaques and epidermal lesions in psoriasis; a chance for dressing development in specific area” by research group Sumate Ampawong. This article is interesting and can be an important contribution to the scientific literature. The reviewer suggests a minor revision is needed before publication in a peer-reviewed journal. If the authors improve the current text of the manuscript, it can be recommended for publication. Some specific comments on the manuscript:

1. Originality

What is the role of alcohol, are there similar results for other types of alcohol?

2. Abstract

Uncontagious immune-mediated skin disease known as psoriasis is regarded as a chronic skin condition with a 10% global prevalence. Prevalence needs to be presented in range, according to WHO.

3. Methodology:

*The authors need to explain methodologies in detail and mention the data sets collected.

4. Results and discussion:

*Present the results with scientific clarity and mechanism of action and make some illustrations for results.  

*State advantages of materials, and state side effects, if any, in the section.

*Suggest improving flow, English. The overall presentation is good, improving readability.

*Specify the limitations of materials at the end of the section.

5. Conclusion

*Ensure conclusions reflect research work results.

*Text is insufficient.  

*Focus on the scope (such as future implications, cost-effectiveness, and how to improve performance)
